# Tunable MEMS-Based Terahertz Metamaterial for Pressure Sensing Application

**DOI:** 10.3390/mi14010169

**Published:** 2023-01-09

**Authors:** Wei-Hsi Lai, Binghui Li, Shih-Huai Fu, Yu-Sheng Lin

**Affiliations:** School of Electronics and Information Technology, Sun Yat-sen University, Guangzhou 510006, China

**Keywords:** metamaterial, terahertz, split-ring resonator, pressure sensor, flow rate sensor

## Abstract

In this study, a tunable terahertz (THz) metamaterial using the micro-electro-mechanical system (MEMS) technique is proposed to demonstrate pressure sensing application. This MEMS-based tunable metamaterial (MTM) structure is composed of gold (Au) split-ring resonators (SRRs) on patterned silicon (Si) substrate with through Si via (TSV). SRR is designed as a cantilever on the TSV structure. When the airflow passes through the TSV from bottom to up and then bends the SRR cantilever, the SRR cantilever will bend upward. The electromagnetic responses of MTM show the tunability and polarization-dependent characteristics by bending the SRR cantilever. The resonances can both be blue-shifted from 0.721 THz to 0.796 THz with a tuning range of 0.075 THz in transverse magnetic (TM) mode and from 0.805 THz to 0.945 THz with a tuning range of 0.140 THz in transverse electric (TE) mode by changing the angle of SRR cantilever from 10° to 45°. These results provide the potential applications and possibilities of MTM design for use in pressure and flow rate sensors.

## 1. Introduction

Metamaterial is an artificial composite material that possesses unique electromagnetic properties [1,2]. The purpose of metamaterial is to break through the limitation of conventional nature materials. For instance, the metamaterial can transform its physical characteristics and then modulate the incident light. The incident electromagnetic wave will be coupled and interacted with metamaterial to show the negative refractive index [3,4], wavefront manipulation, perfect absorption, and phase modulation, which are widely used in many optoelectronics applications [5]. These advantages prompt it to be extensively adapted for visible light [6,7,8], infrared (IR) light [9,10,11,12,13,14], terahertz (THz) frequency [15,16,17,18], etc. Therefore, metamaterials have been reported in many types of research and investigations in couplers [19], antennas [20,21,22], metalens [23,24], programmable logic devices [25,26,27], and sensors [28,29,30] fields. In addition, THz wave is known as far IR ray, which possesses the preponderances of high penetrability, low photon energy, and great coherence [31,32]. The combination of THz wave and metamaterial has commonly been demonstrated in biosensing, chemical gas sensing, and temperature sensing applications [33,34,35]. Among these areas, biological scientists utilize metamaterial-based sensors in the medical and biology field, which can accurately detect the content information of DNA or protein like SARS-CoV-2 [36].

Nonetheless, the introduction of metamaterial into pressure and flow rate sensor is not presented and discussed at present. Compared to the conventional pressure and flow rate sensor, the proposed MTM possesses the characteristics of high accuracy, tunability, and lower energy loss. As a pressure and flow rate sensor application, it should be considered the cost, precision, size, and shape of the device appropriate for the metamaterial design. Recently, the diversified configurations of metamaterials are illustrated and demonstrated [37,38]. Among these configurations, the split-ring resonator (SRR) is the most representative one. As a kind of magnetic and electric metamaterial, SRR is designed as a split metallic loop to form an inductance (L) and capacitance (C) [39]. Assuming there is only one closed metal ring, the electromagnetic field will not form a resonant system. Hence, to induce a resonance and strengthen the magnetic response, a split is required, which is equivalent to a capacitor. This split metallic loop is equivalent to an LC circuit. The charge accumulates at the opening part to generate an electric field and then from a resonant system [40,41].

To make the LC circuit more flexible, manufacturable and applicable, the micro-electro-mechanical system (MEMS) technique has been demonstrated in the literature [42]. It can independently form an intelligent system, which means it does not need to change the material and external parameters of the metamaterial. In this study, we propose a tunable THz metamaterial using the MEMS technique. This proposed MEMS-based THz metamaterial (MTM) is composed of SRR configuration on the patterned silicon (Si) substrate with through Si via (TSV). SRR configuration is released and suspended on TSV, which can be used for pressure sensing applications [43,44,45,46,47,48]. In the initial state, the influences of electromagnetic responses in transverse magnetic (TM) and transverse electric (TE) modes are investigated to optimize the MTM geometric parameters. Second, the electromagnetic responses are discussed when the airflow passes through the TSV from bottom to up and then bends the SRR cantilever. Therefore, the SRR cantilever will be bent upward in a different extent by increasing the airflow rate. Such a design of the proposed MTM could be applied in the high-efficient pressure and flow rate sensors.

## 2. Designs and Methods

Figure 1a shows the schematic drawing of MTM. This design starts from the patterned Si substrate with TSV structures and then designed SRR configuration to form a periodic gold (Au) metamaterial. The key geometric parameters of SRR are width, length, and split defined as *a*, *b*, and *g*, respectively. To optimize the geometrical dimensions of MTM, the key parameters of metamaterial, i.e., the width (*a*), length (*b*), and split (*g*) are compared and discussed. The corresponding geometrical denotations are indicated in Figure 1b. The metallic linewidth is kept as constant as 5 μm and the period is kept as constant as 30 × 30 μm^2^ and the size of TSV is 200 × 200 μm^2^. The specific simulations of electromagnetic responses of MTM are performed by using Lumerical Sololution’s finite-difference time domain (FDTD). The resonant frequency monitor is set below the SRR cantilever to detect the transmission spectra. The propagation direction of incident THz wave is set to be perpendicular to the x-y plane in the simulations. The *x*- and *y*-axis directions are assumed periodic boundary conditions, and the *z*-axis direction is applied in perfectly matched layer (PML) boundary condition. Figure 1b shows the change of the bending angle of the SRR cantilever. When the airflow is passed through the TSV to bend the SRR cantilever, which bending degree can be changed from the initial angle (*θ*) to the bending angle (Δ*θ*) and then tuned the resonance of SRR. By increasing the airflow rate passed through the TSV from bottom to up, the pressure is increased to bend the SRR cantilever and then the bending degree of the SRR cantilever can be changed correspondingly. This mechanical phenomenon can result in the resonant frequency shift of MTM.

## 3. Results and Discussions

The physical principle of SRR can be introduced as Equation (1). The SRR structure can induce the equivalent circuit composed of inductance (L) and capacitance (C) by illuminating the incident THz wave on the MTM device surface. The incident THz wave could be interacted with the SRR structure and then generate a resonant frequency, which corresponding electromagnetic response can be expressed by [49,50,51]
(1)$$\omega_{LC}=2\pi f_{LC}=\frac{1}{\sqrt{L\prime C\prime }}\propto \left(\frac{c_{0}}{s\sqrt{\varepsilon_{c}}}\right)\frac{g}{w}$$
where *c*_0_ is the light velocity in vacuum. *L′* and *C′* are referring to equivalent inductance and capacitance, *ε_c_* is the relative permittivity of MTM, *g* is the split of SRR, *w* is the linewidth of SRR, and *s* is the unit cell’s size of SRR, respectively. The transmissivity of the electromagnetic wave can be calculated by the standard Drude Lorentz model followed by Equation (2).
(2)$$T=\frac{4n_{air}n_{sub}n_{EM}^{2}}{{\left(n_{air}n_{sub}+n_{EM}^{2}\right)}^{2}}$$
where *n_air_* and *n_sub_* are the refractive index of air and substrate, respectively. The refractive index of Si substrate is 3.416. The permittivity of the Au material is described using the Drude model as expressed by [52]
(3)$$\varepsilon \left(\omega \right)=1-\frac{\omega_{p}^{2}}{\omega \left(\omega +i\omega_{c}\right)}\;$$
where $\omega_{p}^{\;}$ = 1.37 × 10^16^ Hz is the plasmon frequency and $\omega_{c}$ = 4.08 × 10^13^ Hz is the scattering frequency for the Au material. It is clear to observe that geometric parameters of MTM (*a*, *b*, and *g*) can influence the resonant frequency of MTM as illustrated in Equations (1) and (2) [53].

Figure 2 shows the transmission spectra of MTM with different *a*, *b*, and *g* values in TM and TE modes. The transmission spectra of MTM by changing *a* value from 14 μm to 30 μm in TM and TE modes as shown in Figure 2a,d, respectively, while keeping *b* and *g* values as constant as *b* = 26 μm and *g* = 12 μm. The resonances are red-shifted from 1.080 THz to 0.810 THz in TM mode and from 0.810 THz to 0.610 THz in TE mode. These trends are quite linear. The resonant spectrum has the strongest response mode and the largest amplitude of resonance in the range of 14 μm to 30 μm. Thus, the optimized *a* value is defined as 26 μm in this study. Figure 2b,e show the transmission spectra of MTM by changing *b* value from 16 μm to 30 μm in TM and TE modes, respectively, while keeping *a* and *g* values as constant as *a* = 26 μm and *g* = 12 μm. The resonances are red-shifted from 1.041 THz to 0.810 THz in TM mode and from 0.760 THz to 0.610 THz in TE mode. These trends are also quite linear. The resonant spectrum has the strongest response mode and the largest amplitude of resonance in the range of 16 μm to 30 μm. The optimized *b* value is defined as 26 μm in this study. Figure 2c,f show the transmission spectra of MTM by changing *g* value from 6 μm to 18 μm in TM and TE modes, respectively, while keeping *a* and *b* values as constant as *a* = *b* = 26 μm. The resonances are blue-shifted from 0.820 THz to 0.890 THz in TM mode and from 0.624 THz to 0.653 THz in TE mode. These trends are also quite linear. The resonant spectrum has the strongest response mode and the largest amplitude of resonance in the range of 6 μm to 18 μm. The optimized *g* value is defined as 12 μm in this study.

To further investigate the physical mechanism of electromagnetic responses of MTM, the electric (E) field and magnetic (H) field energy distributions of MTM with different *a*, *b,* and *g* parameters in TM and TE modes are plotted in Figure 3 and Figure 4, respectively. Figure 3a illustrates the E-field energy distribution under the conditions of *a* = 26 μm, *b* = 30 μm, and *g* = 12 μm. The resonance is at 0.856 THz. The E-field energy distribution is concentrated on the vertexes of SRR, while the H-field energy distribution is gathered within the inside of SRR as shown in Figure 3d. Thus, there is an induced electric current in the SRR that can be equivalent to an LC oscillation circuit. This induced resonance is called LC resonance. In Figure 3b,e, the SRR parameters are *a* = 30 μm, *b* = 26 μm, and *g* = 12 μm, which can induce three equivalent electric currents focused on the contour of SRR. It is obvious to observe that there is an energy concentration difference from the E- and H-field distributions, which can form a cyclic current inside the SRR. This circuit is composed of three capacitors and one inductor to generate a LC oscillation mode. Figure 3c,f are the E- and H-field distributions of SRR at the resonant frequency of 0.821 THz with *a* = *b* = 26 μm and *g* = 6 μm. The E-field energies are mainly concentrated on the inner corner of the opposite gap and the inside of SRR, while the H-field distribution is gathered in the inner opposite SRR.

In Figure 4a,d, the physical mechanism of resonances in TE mode is similar to that in TM mode under the conditions of *a* = 26 μm, *b* = 30 μm, and *g* = 12 μm. The observed field energy distributions are mainly concentrated on the inside of SRR and slightly gather within the corner of the opposite gap. The H-field energy distributions are collected in the inside corner of SRR. By changing *b* value to 26 μm and keeping *a* = 30 μm and *g* = 12 μm, the results of equivalent induced electric currents are shown in Figure 4b,e, which are the same equivalent circuit mode as shown in Figure 4a,d, respectively. Figure 4c,f show the results of E-field and H-field energy distributions under the conditions of *a* = *b* = 26 μm and *g* = 6 μm, which show similar electromagnetic characteristics in TM mode. It can be verified that the change of geometric parameters of SRR will cause the resonant frequency shift.

Figure 5a shows the transmission spectra of MTM with different *θ* values. By increasing the bending degrees of the MTM from 10° to 45°, the resonance will be linearly blue-shifted from 0.721 THz to 0.796 THz with a tuning range of 0.075 THz in TM mode. Figure 5b shows the relationship of *θ* values and resonances. The resonance shift exhibits a linear tendency regarding to *θ* variations. The linearity is 0.997. The E- and H-field energy distributions in TM mode with *a* = *b* = 26 µm, *g* = 12 µm and *θ* = 10° are plotted in Figure 5c,d, respectively. According to the induced electric circuit corresponding to the energy distribution as discussed above, the E-field energy is mainly gathered within the opposite gap and H-field energy is concentrated on the inside surface of SRR. There is an equivalent LC resonant circuit with one capacitor and one inductor induced by an electromagnetic field.

Figure 6a shows the transmission spectra of MTM with different *θ* values. By increasing the bending degrees of the MTM from 10° to 45°, the resonance will be linearly blue-shifted from 0.805 THz to 0.945 THz with a tuning range of 0.140 THz in TE mode. Figure 6b shows the relationship of *θ* values and resonances. The resonance shift also exhibits a linear tendency regarding to *θ* variations. The linearity is 0.983. The E- and H- field energy distributions in TE mode with *a* = *b* = 26 µm, *g* = 12 µm and *θ* = 10° are shown in Figure 6c,d, respectively. According to the induced electric circuit corresponding to the energy distributions, the E-field energy is mainly gathered within the opposite gap and the H-field energy is concentrated on the inside corner of SRR. There is an equivalent LC resonant circuit with one capacitor and one inductor. The great linear trend shown in Figure 5b and Figure 6b can be verified that the increasing bending degrees possess the great tunability for the design of pressure and flow rate sensing applications.

Since the transmission spectra of MTM exhibit an appropriate tendency of *θ* variation, the magnitude of the equivalent updraft by using the force analysis of the cantilevers and the physical and mathematical models can be calculated distinctly. The model can be regarded as a cantilever with the single-clamped beam. The cantilever can be bent in different angles by an upward airflow passing through the TSV. The rising airflow provides the input pressure that makes the cantilever with a homogeneous force. By utilizing the infinitely subdivided single-clamped beam model into small pieces, conducting force analysis and integration on each small piece of the single-clamped beam. The magnitude of the updraft force can be expressed by *F = mgcosθ*. Therefore, the force detected by the cantilever can be calculated by the mass equation, e.g., *m* = *ρ*·*V*. The Au material density is 19.32 g/cm^3^, and the volume can be calculated from the cantilever. Figure 7a shows the relationship between cantilever bending angles and induced bending forces. Owing to the airflow force vertically detected by the MTM cantilever, the MTM cantilever with a smaller bending angle will sustain large airflow upward pressure. Therefore, the MTM cantilever with the smaller bending angle will be detected the increased updraft with a large airflow force. Conversely, when the bending angle of the cantilever becomes large, the contact area of the cantilever by the airflow is smaller, which will cause the detected force smaller. It can be confirmed that the airflow force will result in the resonant frequency shift according to the relationship between the cantilever bending angle and the induced bending forces in Figure 7a, and the relationship between the bending angle and the resonant frequency shift in Figure 5b and Figure 6b, respectively. Figure 7b summarizes the relationships between the resonant frequency and the induced bending force in TM and TE modes. The resonances are red-shifted from 0.796 THz to 0.721 THz with a tuning range of 0.075 THz in TM mode and from 0.945 THz to 0.805 THz with a tuning range of 0.140 THz in TE mode, respectively, by enlarging the induced bending force from 3.1 pN to 4.4 pN. The trends are quite linear with linearities of 0.975 and 0.992 for TM and TE modes, respectively. Additionally, according to the simulation and calculation results, the sensitivity of the MTM device is 0.0591 THz/pN and 0.109 THz/pN for sensing pressure in TM and TE modes, respectively, and 13.727 THz/(m/s)^2^ and 24.739 THz/(m/s)^2^ for sensing airflow rate in TM and TE modes, respectively. Such results can appropriately reflect that the MTM design provides an effective approach to detecting the change of the airflow and the induced bending force of MTM and then performing the sensing application.

## 4. Conclusions

In conclusion, a MTM device is presented to be used as pressure and airflow rate sensing applications. By increasing the bending degree of the cantilever from 10° to 45° to determine the pressure of the external airflow passing through the TSV. The magnitude of the induced bending forces is from 4.4 pN to 3.1 pN, while that of the resonant frequency can be tuned from 0.721 THz to 0.796 THz with a tuning range of 0.075 THz in TM mode and from 0.805 THz to 0.945 THz with a tuning range of 0.140 THz in TE mode, respectively. Both of TM- and TE-polarization resonances are blue-shifted by increasing the bending degree of the cantilever. The sensitivity of the MTM device is 0.0591 THz/pN and 0.109 THz/pN for sensing pressure in TM and TE modes, respectively, and 13.727 THz/(m/s)^2^ and 24.739 THz/(m/s)^2^ for sensing airflow rate in TM and TE modes, respectively. These results can indicate that MTM can potentially be used as pressure and airflow rate sensing applications. The proposed MTM design can easily realize the merits of high sensitivity, high integration, and high accuracy for sensing applications. Such pressure and airflow rate sensing applications provide excellent contributions to the different fields of scientific research, such as biological, liquid, environmental, and humidity sensors.

## Figures and Tables

**Figure 1 micromachines-14-00169-f001:**
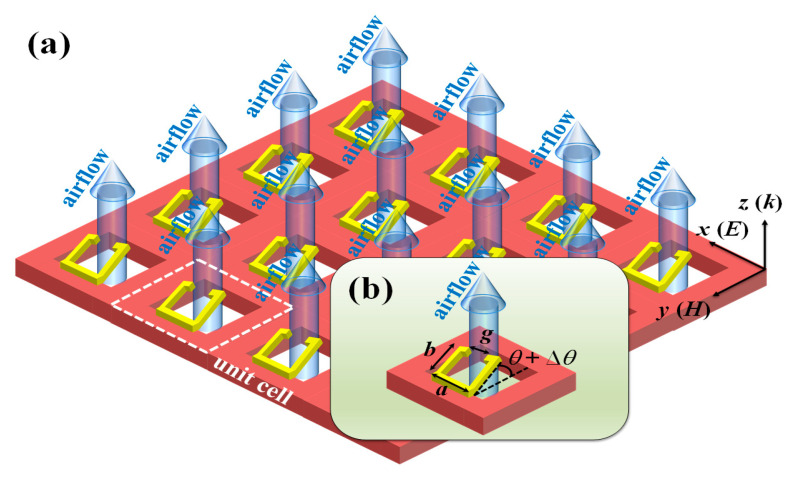
Schematic drawings of 3D MTM (**a)** array and (**b**) unit cell for pressure sensing application. The geometric parameters of MTM are width (*a*), length (*b*), split (*g*), and the tilted angle of the cantilever (*θ*) of SRR.

**Figure 2 micromachines-14-00169-f002:**
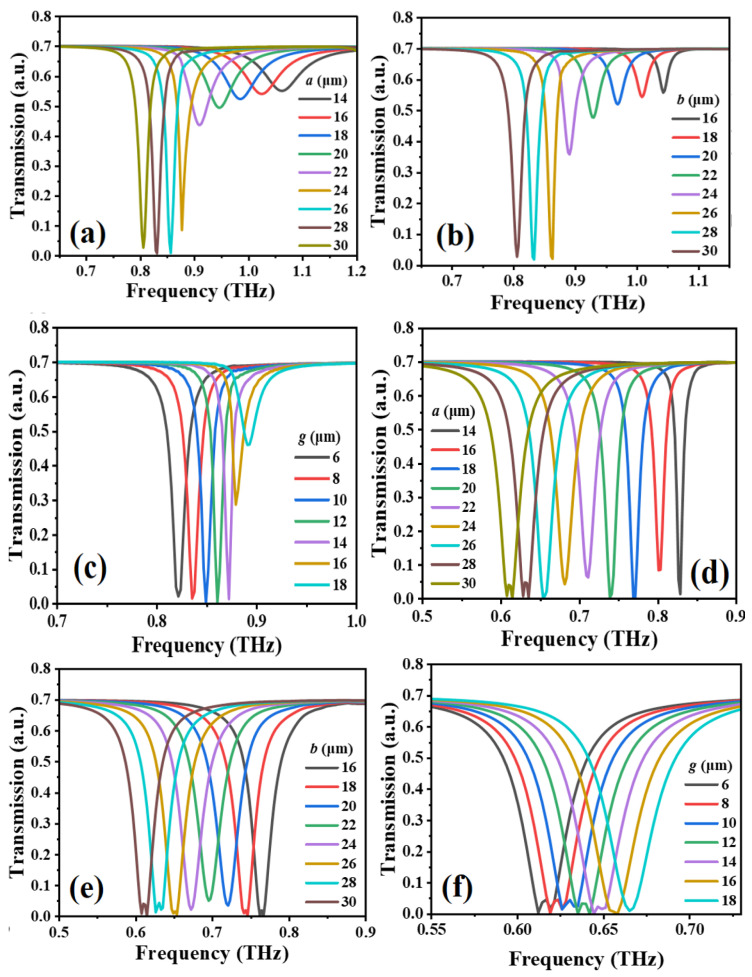
Transmission spectra of MTM with different (**a**,**d**) *a*, (**b**,**e**) *b*, (**c**,**f**) *g* values in (**a**–**c**) TM and (**d**–**f**) TE modes.

**Figure 3 micromachines-14-00169-f003:**
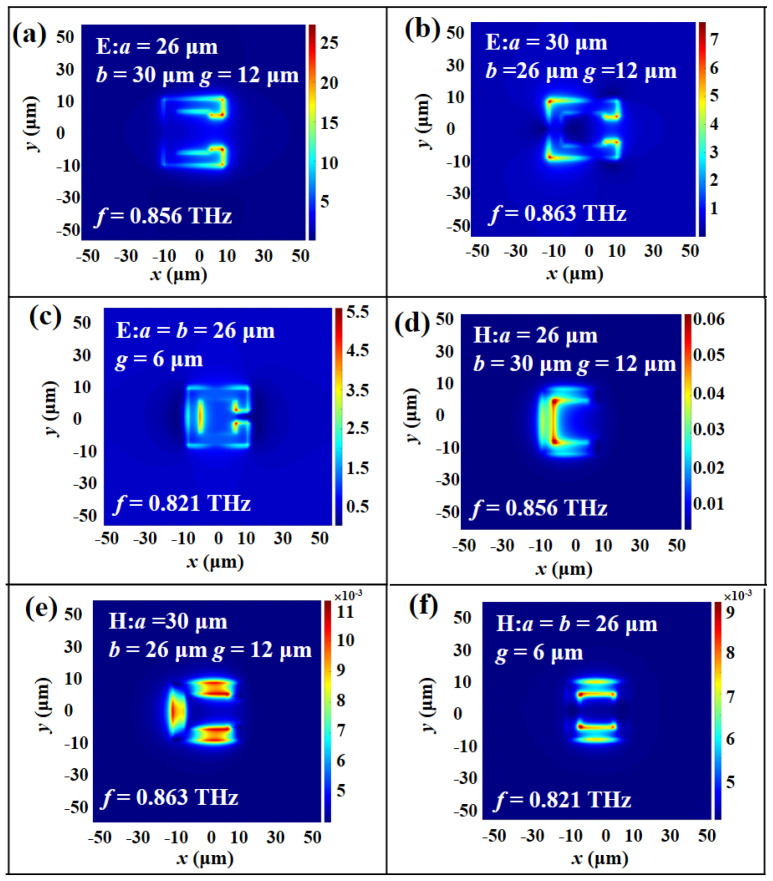
(**a**–**c**) E– and (**d**–**f**) H–field distributions of MTM at the condition of *a*, *b*, and *g* values in TM mode.

**Figure 4 micromachines-14-00169-f004:**
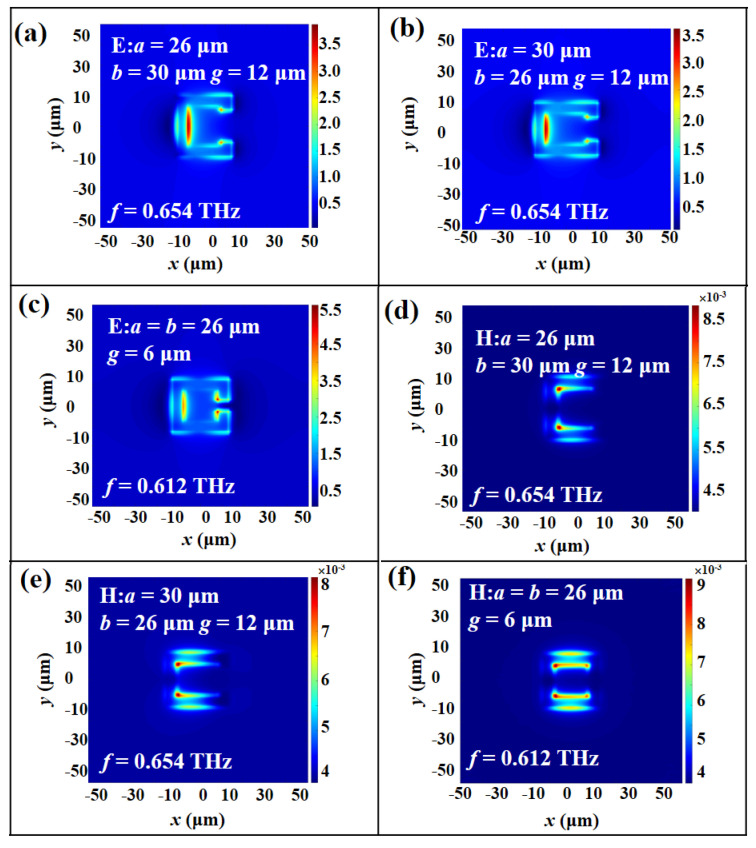
(**a**–**c**) E– and (**d**–**f**) H–field distributions of MTM at the condition of *a*, *b*, and *g* values in TE mode.

**Figure 5 micromachines-14-00169-f005:**
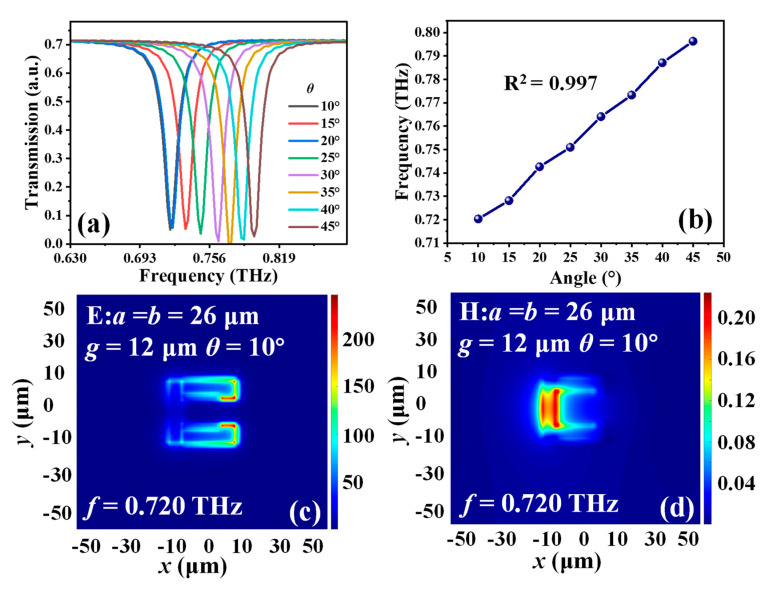
(**a**) Transmission spectra of MTM with different *θ* values. (**b**) Relationship of *θ* values and resonances. (**c**) E– and (**d**) H–field distributions of MTM with *a* = *b* = 26 µm, and *θ* = 10° in TM mode.

**Figure 6 micromachines-14-00169-f006:**
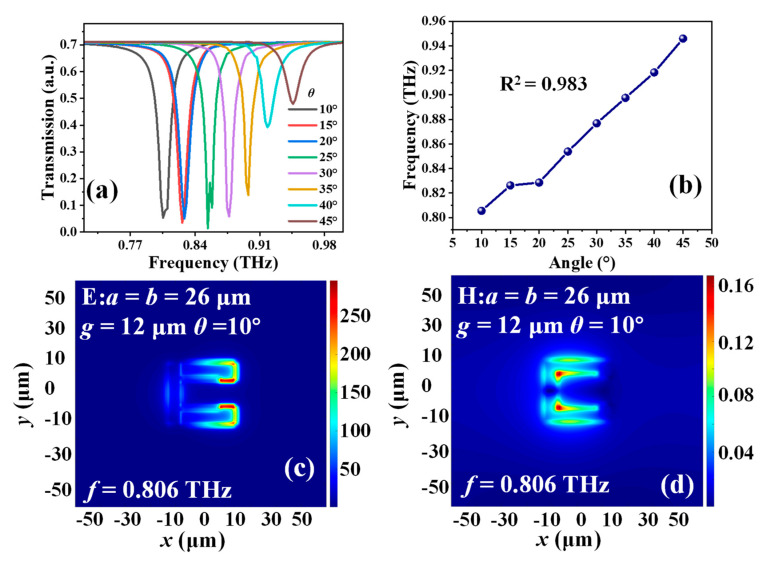
(**a**) Transmission spectra of MTM with different *θ* values. (**b**) Relationship of *θ* values and resonances. (**c**) E– and (**d**) H–field distributions of MTM with *a* = *b* = 26 µm, and *θ* = 10° in TE mode.

**Figure 7 micromachines-14-00169-f007:**
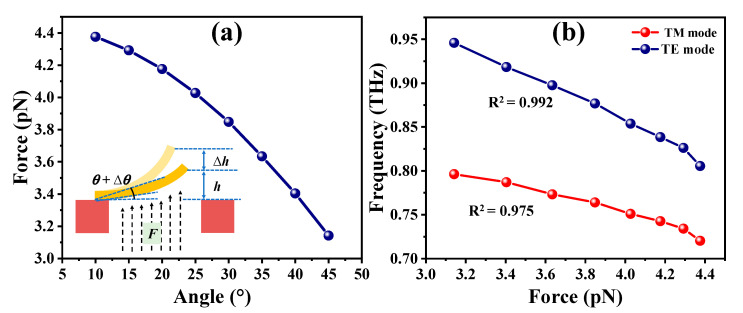
(**a**) Relationship of cantilever bending angles and induced bending forces. (**b**) Relationships of resonances and induced bending forces of MTM in TM and TE modes.

## Data Availability

Not applicable.

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
