# Peer review of "Tunable MEMS-Based Terahertz Metamaterial for Pressure Sensing Application"

_micromachines, 2023, doi:10.3390/mi14010169_

Round 1

Reviewer 1 Report

The article discusses the possibilities of using the MTM structure as pressure sensing device. It is shown that the resonant frequency of the structure changes under the action of an external force. Unfortunately, it requires a bulky and expensive THz spectrometer, making its practical application highly questionable in the near future.

1. The work is pure theoretical (maybe I’m not right) so me confuses the Acknowledgments in which authors thanks laboratory for the use of experimental equipment.

2. Bending at high angles of cantilever could produce plastic deformation so the sensor becomes useless. If authors did not agree with this they must provide arguments.

3. MTM structure is proposed as a pressure and/or air flow sensor, but the article lacks any information about pressure and air flow. A value of the article would increase if the comments on the sensitivity of the device (minimum pressure, air flow it could detect) and the range of pressures it could register were added.

Author Response

Reviewer #1:

The article discusses the possibilities of using the MTM structure as pressure sensing device. It is shown that the resonant frequency of the structure changes under the action of an external force. Unfortunately, it requires a bulky and expensive THz spectrometer, making its practical application highly questionable in the near future.

  1. The work is pure theoretical (maybe I’m not right) so me confuses the Acknowledgments in which authors thanks laboratory for the use of experimental equipment.

Response:

Thanks for the reviewer's kindly instruction. We have revised the acknowledgment as below.

“…for the use of simulation code.”

  1. Bending at high angles of cantilever could produce plastic deformation so the sensor becomes useless. If authors did not agree with this they must provide arguments.

Response:

Thanks for the reviewer's kindly instruction. The cantilever structure is Au material, that possesses bendability and ductility. Thus, it can ignore the plastic deformation and directly use the lever model to analyze and calculate it.

  1. MTM structure is proposed as a pressure and/or air flow sensor, but the article lacks any information about pressure and air flow. Avalue of the article would increase if the comments on the sensitivity of the device (minimum pressure, air flow it could detect) and the range of pressures it could register were added.

Response:

Thanks for the reviewer's kindly instruction. Additionally, according to the simulation and calculation results, the sensitivity of the MTM device is 0.0591 THz/pN and 0.109 THz/pN for sensing pressure in TM and TE modes, respectively, and 13.727 THz/(m/s)2 and 24.739 THz/(m/s)2 for sensing airflow rate in TM and TE modes, respectively.

Reviewer 2 Report

The authors present a tunable THz metamaterial to demonstrate pressure and flow rate sensing applications. The proposed device is based SRR as a cantilever on TSV structure. When the air flow passes through the TSV from bottom to up, which will be bent the SRR cantilever upward and then tune the corresponding resonance. The manuscript has a fair to good standing with respect to technical content. It is an interesting work. I am very happy to recommend its publication with some minor concerns listed as below.

1.        The authors should provide more detail settings of the software and method applied in the simulations. In addition, can the authors give some more details about the Au and the permittivity of substrate in numerical simulation? It is not clear how the authors model these materials in simulations. Since the study is purely theoretical, the authors should give more information about the simulation. 2.        The author only gives the range of parameter changes, and does not give the specific parameters used for each transmission spectrum. I cannot know the parameter settings for each simulation figure. For example, what are the parameters of TSV? 3.        The paper lacks of interpretation of the observations made from the energy field distributions. Authors sometimes just report their observation without any explanation. Is the effect obtained in the manuscript a plasma-induced transparency effect? The authors should mark the frequency of the energy field distributions to be analyzed in the corresponding transmission spectra for better readability of the paper. Why do the authors analyze the field distributions at these frequencies? Please explain in the manuscript.

4.        The authors are also suggested to check the grammar and language of the manuscript.

5.        Considering previous works have been done in this field, the authors are suggested to give credits to them. Here I list several references for consideration:

Ÿ   iscience, 103799, 2022.

Ÿ   Int. J. Optomechatron. 15, 120 - 159, 2021.

Ÿ   Int. J. Optomechatron. 15, 97 - 119, 2021.

Ÿ   Int. J. Optomechatron. 14, 78 - 93, 2020.

Ÿ   Int. J. Optomechatron. 16, 42 - 57, 2022.

Round 2

Reviewer 1 Report

I am satisfied with the answers.